# A Novel Aldisine Derivative Exhibits Potential Antitumor Effects by Targeting JAK/STAT3 Signaling

**DOI:** 10.3390/md21040218

**Published:** 2023-03-29

**Authors:** Dong-Ping Wang, Li-Hong Wu, Rui Li, Na He, Qian-Yue Zhang, Chen-Yang Zhao, Tao Jiang

**Affiliations:** 1Key Laboratory of Marine Drugs, Ministry of Education, School of Medicine and Pharmacy, Ocean University of China, Qingdao 266003, China; 2Laboratory for Marine Drugs and Bioproducts, Qingdao National Laboratory for Marine Science and Technology, Qingdao 266237, China; 3Innovation Platform of Marine Drug Screening & Evaluation, Qingdao National Laboratory for Marine Science and Technology, Qingdao 266100, China; 4Department of Cancer Biology, Cleveland Clinic, 9500 Euclid Ave, Cleveland, OH 44195, USA

**Keywords:** JAK/STAT3 signaling pathway, aldisine derivatives, isothiouronium, antitumor activity, JAK inhibitor, high-throughput screening

## Abstract

The JAK/STAT3 signaling pathway is aberrantly hyperactivated in many cancers, promoting cell proliferation, survival, invasiveness, and metastasis. Thus, inhibitors targeting JAK/STAT3 have enormous potential for cancer treatment. Herein, we modified **aldisine** derivatives by introducing the isothiouronium group, which can improve the antitumor activity of the compounds. We performed a high-throughput screen of 3157 compounds and identified compounds **11a**, **11b**, and **11c**, which contain a pyrrole [2,3-c] azepine structure linked to an isothiouronium group through different lengths of carbon alkyl chains and significantly inhibited JAK/STAT3 activities. Further results showed that compound **11c** exhibited the optimal antiproliferative activity and was a pan-JAKs inhibitor capable of inhibiting constitutive and IL-6-induced STAT3 activation. In addition, compound **11c** influenced STAT3 downstream gene expression (Bcl-xl, C-Myc, and Cyclin D1) and induced the apoptosis of A549 and DU145 cells in a dose-dependent manner. The antitumor effects of **11c** were further demonstrated in an in vivo subcutaneous tumor xenograft experiment with DU145 cells. Taken together, we designed and synthesized a novel small molecule JAKs inhibitor targeting the JAK/STAT3 signaling pathway, which has predicted therapeutic potential for JAK/STAT3 overactivated cancer treatment.

## 1. Introduction

Cancer presenting as an incurable advanced or metastatic disease is common, and the development of new targeted antitumor drugs is of great significance to human health [1]. The Janus kinase (JAK) and signal transducer and activator of transcription 3 (STAT3) signaling pathway is aberrantly hyperactivated in many types of cancer, and such hyperactivation is generally associated with a poor clinical prognosis [2,3]. It plays a central role in immune response, cell proliferation, differentiation, and survival [4,5]. In the tumor microenvironment, numerous cytokines, such as IL-6 and IFNα/β, can activate JAKs by phosphorylating tyrosine residues, which in turn phosphorylate and activate STAT3 to regulate the transcription of downstream target genes [6,7]. Examples of these genes include those which drive cell proliferation (cyclin D1), promote tumor survival BCL2-like protein 1 (BCL-xL), and regulate immune response (IFN-γ) [2,8]. There are 11 JAK inhibitors approved for the treatment of various diseases that exert inhibitory effects through competitive and non-competitive reactions with the amino acid residues in JAKs [9]. An example of a JAK inhibitor used in treatment is ruxolitinib, which has been approved by the U.S. Food and Drug Administration (FDA) for the treatment of myelofibrosis and polycythemia vera [10]. In addition, dozens of potential JAK inhibitors are under clinical trials to further evaluate their antitumor activity; for example, DZD4205 in the treatment of T-cell lymphoma [11]. Therefore, the JAK/STAT pathway is a potential therapeutic target with great value for cancer treatment; among these, small molecules to inhibit JAK activity have attracted much attention.

Marine sponges are an important source of potential natural bioactive ingredients that have formed natural secondary metabolites with complex and unique structures to adapt to the harsh marine survival conditions [12,13]. **Aldisine** (Figure 1) and its derivatives are the secondary metabolites from sponges with unique pyrrolo [2,3-c] azepine skeleton structural features that exhibit antitumor activity by inhibiting protein kinase [14,15]. It was reported that the oximes, oxime ethers, and hydrazones groups in **aldisine** can provide numerous hydrogen bond donor and acceptor moieties, which showed antiviral, larvicidal, and anti-phytopathogenic-fungus activities [16]. Modified **aldisine** at N-1 and C-4 positions obtained compounds **1** to **3** (Figure 1), which could enhance its antiproliferative activity [17,18,19,20]. The extensive antiproliferative properties of **aldisine** have made it a significant drug for cancer therapy. Due to its promising biological activities and multi-site modification, **aldisine** has become a focus of pharmaceutical chemistry as a lead compound, especially in antitumor aspects.

These previous results guide us to put our efforts into designing and modifying **aldisine**. Isothiouronium is a positively charged group that induces G2/M cell cycle arrest and promotes cell apoptosis [21,22]. Previous studies have shown that isothiouronium-modified analogs may be promising anticancer agents, novel Golgi staining reagents, and useful research tools for studying Golgi functions in normal or cancer cells [23]. Therefore, compounds **10a**, **10b**, **10c**, **11a**, **11b**, and **11c** were obtained by introducing the isothiouronium groups to **aldisine** at the N-1 and N-7 positions with different lengths of carbon alkyl chains. The carbon alkyl chains increase their flexibility and liposolubility, and the introduction of isothiouronium makes it easier to enter cells to enhance antitumor activity.

Therefore, we first tested the antitumor activities of **aldisine** and its derivatives and found that compounds **11a**, **11b**, and **11c** significantly reduced cell viability in the treatment of cancer cells. In parallel, we screened tens of thousands of collected compounds in a STAT3 binding promoter driving luciferase reporter system according to the method reported in references [20,21], in which compounds **11a**, **11b**, and **11c** showed remarkable STAT3 inhibitory activity, with compound **11c** showing the optimal activity. Further experiments focused on compound **11c** revealed its inhibitory efficacy against JAKs. In addition, inhibition of JAK/STAT3 signaling by compound **11c** induced substantial tumor cell apoptosis and influenced cell proliferation. Notably, compound **11c** significantly reduced tumor growth by induced apoptosis in a mouse subcutaneous tumor implantation model in vivo, suggesting a potential for compound **11c** to be further developed as a JAK/STAT3 signaling inhibitor for the treatment of cancer.

## 2. Results

### 2.1. Chemistry

#### Design and Synthesis of Aldisine Derivatives

Considering the rotatability of different lengths of carboxyalkyl chains (*n* = 4,5,6) and the positive charge of the isothiouronium group, which can easily approach the cell membrane, we first designed and synthesized isothiouronium-modified **aldisine** derivatives at N1 positions in two steps as outlined in Figure 1. Compound **7** was prepared from the 2-(trichloroacetyl) pyrrole according to the synthesis procedure of marine natural products **Stevensine** [24]. Subsequently, compound **7** reacts with excessive dibromoalkanes (*n* = 4,5,6) in DMSO for 1.5 h under the basic condition of KOH to obtain bromoalkyl derivatives **8a**, **8b**, **8c**, **9a**, **9b**, and **9c**, which were separated by silica gel chromatography with a 39~56% yield. Then **8a**, **8b**, **8c**, **9a**, **9b**, and **9c** were reacted with thiourea reflux for 12 h in ethanol to produce the target compounds **10a**, **10b**, **10c**, **11a**, **11b**, and **11c** with a yield of 57~67%.

The general synthesized procedures for compounds **8a**, **8b**, **8c**, **9a**, **9b**, **9c** and **10a**, **10b**, **10c**, **11a**, **11b**, **11c** are in the Appendix A.

### 2.2. Biological Activity Assessment of Compound **11c**

#### 2.2.1. Compound **11c** Exhibited Antiproliferative Activity and was Identified as a JAK-STAT3 Signaling Inhibitor

After a series of **aldisine** derivatives were synthesized with similar structures and chemical properties, as shown in the chemistry section, we conducted high-throughput screening on them and found that the compound **11c**, containing two six-carbon chain lengths of alkyl isothiourea groups at the N1 and N7 positions significantly inhibit JAK/STAT3 activity. Whereas compounds **10a**, **10b**, and **10c**, containing only one different length of alkyl isothiourea at the N1 position, demonstrated lower inhibition of JAK/STAT 3 activity than **11c** (Appendix A).

Next, for further understanding of the structure-activity relationship between **aldisine** derivatives, the inhibitory activities of **11a**, **11b**, and **11c** were tested. The results showed that compounds **11a**, **11b**, and **11c** exhibited significant antiproliferative activity against four human cancer cell lines: prostate cancer (DU145), non-small cell lung cancer (A549), breast cancer (MDA-MB-231), and cervical cancer (HeLa) (Table 1), with **11c** being the most potent. Compound **11c** inhibited the growth of DU145 cells with an IC_50_ value of 2.37 μM (Figure 2A) and HeLa cells with an IC_50_ value of 5.12 μM (Figure 2B). STAT3 can be activated in cancer cells in a constitutive or IL-6-induced manner [25,26]. In both DU145 and A549 cells, STAT3 is constitutively activated, whereas in Hela and MDA-MB231 cells, IL-6 can induce significant STAT3 activation [27,28,29]. Therefore, we next evaluated whether **aldisine** derivatives can inhibit STAT3 signaling. A STAT3 transcriptional activity-based high-throughput luciferase reporter was applied [23]. Interestingly, STAT3 luciferase inhibitory activities as well as the antiproliferative effects were indeed observed for **11a**, **11b**, and **11c** (Appendix A; Figure 2C,D). The IC_50_ value of **11a**, **11b**, and **11c** was generally lower and exhibited greater inhibitory activity as the length of alkyl carbon groups increased. Compound **11c** with two six-carbon chain lengths showed the highest activity, which inhibited cancer cell growth in a dose-dependent manner and could be a potential JAK/STAT3 pathway inhibitor at a relatively low concentration.

#### 2.2.2. **11c** Inhibits Constitutive and IL-6-induced STAT3 Activation

To further determine whether **11c** can inhibit the activation of STAT3, we first treated DU145 and A549 cells with **11c** at the indicated concentrations for 2 h and then analyzed the phosphorylation of STAT3 by Western blot (Figure 3A,B). Compound **11c** inhibited constitutively activated STAT3 at the concentration of 5 μM both in DU145 and A549 cell lines. In consideration of IL-6 as the major cytokine participant in STAT3 activation and its significance in cancer formation and suppressing the antitumor immune response [4,30], we further examined the effect of **11c** on IL-6-induced STAT3 activation cell lines such as Hela and MB231 cells (Figure 3C,D). The results indicated that **11c** inhibited non-constitutive STAT3 activation at the concentration of 7 μM in a dose-dependent manner. Taken together, the Western blot analysis results suggested that **11c** treatment inhibits the constitutive and IL-6-induced STAT3 phosphorylation in cancer cells.

#### 2.2.3. **11c** Inhibits the Phosphorylation of JAK Family Members

The activation of STAT3 is usually regulated by its upstream JAK kinases phosphorylation [31]. To further explore whether **11c** inhibits STAT3 phosphorylation by influencing JAK kinases, DU145 and A549 cells were treated with **11c** for 1 h and JAK kinase phosphorylation was determined (Figure 4A,B). Phosphorylation of JAK1, JAK2, JAK3, and TYK2 were decreased after **11c** treatment at 5 μM in DU145 and 7.5 μM in A549 cells. These data suggest that **11c** is a pan-JAK inhibitor and has a higher affinity. Then, to determine whether **11c** specifically inhibited the JAK/STAT3 pathway or not, we analyzed other signaling kinases including IKK, AKT, NF-κB, p38, p-JNK, and GSK-3β (Figure 4C,D). The results indicated that **11c** had no substantial inhibitory effect on the phosphorylation of IKK, AKT, and GSK-3β, with slight inhibitory activities on NF-κB at the higher concentration and stimulatory effects on the phosphorylation of p38 and JNK. In conclusion, the above data demonstrate that **11c** inhibits the activation of JAK kinases in constitutively activated STAT3 cancer cells.

#### 2.2.4. Molecular Docking Revealed That Hydrogen Bonding Is the Major Interaction between **11c** and JAKs

JAKs, with a total length of 120–140 kD, are essential for many biological outcomes of cytokine signaling. The JAK family contains four members (JAK1, JAK2, JAK3, and TYK2), with the end being a catalytic or kinase domain at the carboxyl end, and the front is a pseudokinase (PK) or kinase-like domain. JAK binds to the Box1 and Box2 domains of cytokine receptors through the amino-terminal ezrin-radixin- moesin (FERM) domain and Src Homology 2 (SH2) domain [32,33,34]. Molecular docking software provides a good platform for the interaction between molecules (ligands) and target proteins (receptors) [32]. In our research, molecular docking was conducted to examine the interaction between compound **11c** and JAK1 (PDB number: 4EHZ), JAK2 (PDB number: 7Q7W), JAK3 (PDB number: 4Z16), and TYK2 (PDB number: 4GJ2). The data indicated that the hydrophobic fatty chain was surrounded by a hydrophobic pocket. In addition, the isothiouronium and hydrogen interact with amino acid residues through hydrogen bonding, forming six hydrogen bonds with Glu-957, Met-956, Gly-1020, and Asp-1039 in JAK1 (Figure 5A). Similarly, in JAK2, it forms four hydrogen bonds with Lys-857, Met-865, and Gln-853 residues (Figure 5B), while in TYK2, it forms two hydrogen bonds with Asp-988 and Tyr-989 (Figure 5D). Specifically, in JAK3, in addition to four hydrogen bonds with Asp-912 and Asp-967, a π–π interaction is also formed with the indole ring of compound **11c** (Figure 5C). Overall, molecular docking revealed that the hydrogen bond is the main interaction between **11c** and JAK members; furthermore, the interaction intensity differences in the four JAK members were related to the inhibition effect of **11c**.

#### 2.2.5. **11c** Downregulates Anti-apoptosis Gene Expression and Induces Cancer Cell Apoptosis In Vitro

Since JAK/STAT3 signaling plays an essential role in cell proliferation and survival by reducing the expression of cell-cycle-related genes (Cyclin D) and anti-apoptotic genes (Bcl-xl) [35,36], we analyzed the effect of **11c** on cell-cycle-related gene expression. As shown in Figure 6A,B, the downstream genes C-Myc, Bcl-xL, and cyclin D1 were significantly downregulated at the **11c** concentration of 7.5 μM and both DU145 and A549. Furthermore, we investigated whether **11c** induced apoptosis in vitro by flow cytometry. As the results show, in DU145 cells, the ratio of cells in early (Annexin V^+^/PI^−^) and late apoptosis (Annexin V^+^/PI^+^) was a total of 19.16% in the 7.5 μM **11c** treatment groups and reached 52.35% in the 10 μM **11c** treatment groups (Figure 6C). Similarly, in A549 cells, the ratio of cells in early and late apoptosis was a total of 20.49% in the 7.5 μM **11c** treatment groups and reached 52.4% in the 10 μM **11c** treatment groups (Figure 6D). Therefore, we demonstrated that **11c** downregulated anti-apoptosis gene expression and induced apoptosis activities in vitro. 

#### 2.2.6. **11c** Inhibits the Growth of DU145 Cells by Inducing Apoptosis In Vivo

To further investigate the effects of **11c** on cancer cell growth in vivo, we examined the effect of compound **11c** in the DU145 xenograft tumors nude model. Tumor xenograft animal models, especially subcutaneous tumor xenograft model in mice, is a widely used tool to bridge basic and clinical cancer research [37,38]. After establishing the model and drug treatment for 21 days, as shown in Figure 7A,B, 10 mg/kg **11c** potently inhibited DU145 cell growth compared to the controlled vehicle group. The tumor inhibitory effects at the 10 mg/kg **11c** (i.p.) dose reached more than 40% efficiency. The solid tumor images after treatment are shown in Figure 7C. In addition, we investigated whether the inhibition of **11c** on tumor growth in vivo also functions via induction of cell-cycle arrest and apoptosis, which is similar to the observation obtained in the in vitro experiments (Figure 6). Ki67, Tunel, and p-STAT3 staining were performed, as shown in Figure 7D, and the Ki67 positive area decreased after **11c** treatment, indicating inhibited tumor proliferation. Meanwhile, an increased Tunel positive area indicated that **11c** treatment caused tumor cell apoptosis in vivo. Consistent with these observations, **11c** inhibited the phosphorylation of STAT3 in vivo. The body weight of the mice (Figure 7E) suggests that all treatments exhibited high efficacy without toxicity.

## 3. Discussion

The JAK/STAT3 is a classical intracellular signaling pathway in the regulation of tumor cell proliferation, survival, invasiveness, and metastasis [2,4,39]. Therefore, the JAK/STAT3 signaling pathway was considered as an effective therapeutic target for numerous cancers [40]. In the current study, we synthesized **aldisine** derivatives named compounds **11a**, **11b**, and **11c**, and their inhibitory effect on JAK/STAT3 signaling was explored in a STAT3 transcriptional activity driven luciferase reporter cell line. In particular, compound **11c** with the longest carbon atom links exhibited the highest biological activity. The Western blot analysis results elucidated that **11c** inhibits the phosphorylation of STAT3 and its upstream JAKs. Compound **11c** was identified as a new pan-JAK kinase inhibitor. Moreover, the introduction of two isothiouronium groups contains a stronger film-breaking property with a positive charge that is superior to a single group. In addition, compound **11c** induced the apoptosis of A549 and DU145 cancer cells by decreasing cell cycle and anti-apoptosis gene expression. Further in vivo antitumor studies showed that apoptosis also plays an important role in **11c** suppressed xenograft tumor growth. Taken together, we have identified a novel compound **11c** that inhibits the phosphorylation of JAKs that induce cell apoptosis to decrease cell proliferation based on the JAK/STAT3 signaling pathway.

The improved understanding of the JAK/STAT3 signaling participation in cancer treatment has led to the increasing discovery of therapeutic intervention with JAK inhibitors [41]. Previous reports have indicated that JAK inhibitors can be used to treat rheumatoid arthritis, cancer, hemophagocytic lymphohistiocytosis, atopic dermatitis, and so on [42,43,44]. Nowadays, several JAK inhibitors have been developed, but the clinical use of pan-JAK inhibitors in cancer therapy is still limited, perhaps due to the selectivity of JAK inhibitors [45,46]. Recent research has focused on more selective JAK inhibitors because specific JAK inhibitors may reduce side effects while increasing safety and efficacy [47,48]. Our molecular docking data showed that the hydrogen bond is the major interaction route between **11c** and the JAK’s amino acid residues. In addition, the hydrogen bond interaction intensity, which was different in the four JAK members, was related to the inhibitory effect of **11c**. This discovery may provide evidence for the further modification of compound **11c** to improve its selectivity of JAK members. At the same time, there are some therapeutic interests in the future; for example, which JAK structural motif the **11c** is targeting and how it works concretely.

In conclusion, our research identified compound **11c** as a novel antitumor drug by specifically targeting the JAK/STAT3 signaling pathway. Compound **11c** exhibited inhibitory activity on JAKs activation, and efficiently inhibited cancer cell growth in vitro and in vivo. The synthesis and modification route of compound **11c** provides the potential for marine drug development in cancer treatment. In the future, compound **11c** may serve as a new skeleton molecule for further development into a JAKs’ inhibitor, which has great potential for cancer treatment in clinical applications.

## 4. Materials and Methods

### 4.1. Chemical Design and Synthesis Materials

All raw materials required in this article were purchased from commercial channels. All reactions were tracked by thin-layer chromatography (TLC) and observed under 254 nm ultraviolet light using precoated silica gel plates. The specification of silica gel used for column chromatography is 200–300 mesh. ^1^H NMR and ^13^C NMR spectra were recorded with a Jeol JNM-ECP 600 MHz spectrometer (Celes Automation Technology Co., Ltd., Tianjin, China), using TMS as an internal standard. Chemical shifts were reported on the δ scale and *J* values were given in Hz. High resolution mass spectra (HRMS) were recorded on a Q-TOF Global Mass Spectrum (Agilent, Beijing, China). The following abbreviations are used: s = singlet, d = doublet, t = triplet, q = quartet, m = multiplet, dd = double-doublet.

### 4.2. Antibodies and Reagents

Antibodies against STAT3, p-STAT3 (Tyr705), JAK1, JAK2, JAK3, TYK2, Bcl-xl, C-Myc, CyclinD1, p-JAK1 (Tyr1022/1023), p-JAK2 (Tyr1007/1008), p-TYK2 (Tyr1054/1055), p-JAK(Tyr980/981), p-IKK-α/β(Ser176/180), p-NF-κB p65(Ser536)(93H1), p-Akt (Ser473) (D9E) XP^®^, p-GSK-3β(Ser9)(D85E12), p-p38 (Thr180/Tyr182), p-JNK (Thr183/Tyr185), and β-Actin were all obtained from Cell Signaling Technology (Danvers, MA, USA). Obtained recombinant human IL6 (Cat. 216-16) was from PeproTech. Protease and phosphatase inhibitors A and B were purchased from Millipore (Billerica, MA, USA). Bioactive drugs and compounds used for high-throughput screening were provided by TargetMol (Shanghai, China). **Aldisine** derivatives were obtained as mentioned in the chemical part.

### 4.3. Cell Culture

A549, DU145, MDA-MB231, and Hela were purchased from the American Type Culture Collection (Manassas, VA, USA). SKA cells were established by transfected A549 cells with a vector containing STAT3-based luciferase reporter gene as in previously reported methods [49]. The DU145 cells were cultured in RPMI 1640 complete medium (Gibco, Grand Island, NY, USA), the others were cultured in DMEM complete medium (Gibco, Grand Island, NY, USA). Prepared complete medium with added 1% streptomycin, 1% penicillin, and 10% fetal bovine serum (FBS, Gibco, Grand Island, NY, USA). Cells were incubated in a 37 °C temperature incubator with 5% CO_2_.

### 4.4. Animals

Nude mice (male, six-week-old, weight 17–20 g) were purchased from Beijing Vital River Laboratory Animal Technology (Beijing, China). Mice were bred in a specific pathogen-free environment with controlled temperature and humidity and 12 h light and dark alternate. The Committee of Experimental Animals of the Ocean University of China (OUC-SMP-2020-11-01) has approved the animal experiments.

### 4.5. Luciferase Reporter Assay

SKA cells (8000 cells/well) were bedded into white 96-well plates (Corning, NY, USA) and incubated in an incubator at 37 °C containing 5% CO_2_ overnight and then treated with the vehicle or indicated concentrations of compounds. Twenty-four hours later, luciferase activity was detected by adding a luciferase substrate (Cat. E2510, Promega, Beijing, China) and read by a SpectraMax^®^ L microplate reader (Molecular Devices, Shanghai, China) [49,50].

### 4.6. Western Blot Analysis

After stimulating with compounds, we washed the cells twice with phosphate-buffered saline (PBS) and lysed the cells with the RIPA buffer including 1% phosphatase and 1% protease inhibitors. Protein lysis was quantified by a BCA kit (Solarbio, Beijing, China) and separated by 10% SDS-PAGE. Later they were transferred onto nitrocellulose membranes (GE Healthcare) at 90v for 2 h and incubated with primary antibodies at 4 °C overnight. After that, they were incubated with anti-rabbit horseradish peroxidase (HRP)-conjugated secondary antibodies at room temperature for 1 h, detected with chemiluminescence HRP substrate (Millipore, Billerica, MA, USA) and photographed by Chemiluminescence Imaging System (Tanon 5200, Shanghai, China).

### 4.7. Cell Viability and Antiproliferation Activity Assay

Cell viability and antiproliferation activity was evaluated by the resazurin indicator [51]. Cells (3500 cells/well) were seeded into 96-well plates overnight and treated with the vehicle or indicated drug concentrations for 72 h. An amount of 10 µL of 1 mg/mL resazurin solution was added per well and, after incubating for 3 h, relative cell viability was detected using a SpectraMax Mode Plate Reader (Molecular Devices, Shanghai, China) at a 595-nm emission wavelength and a 549-nm excitation wavelength. The half-maximal inhibitory concentration value (IC_50_) of **11c** was determined using GraphPad Prism 8 software.

### 4.8. Molecular Docking

The docking results of compound **11c** (ligand) and JAK proteins (receptor) were calculated by the MOE (version 2020) molecular docking program. Crystal structures of all JAK proteins were obtained from the RCSB Protein Data Bank (RCSB PDB). Firstly, compound **11c** was subjected to an energy minimization processing through Chemdraw 3D software (version, 14.0.0.17). Secondly, the water molecules of the receptor protein and the existing small molecule ligands in the crystal were removed. Finally, compound **11c** was introduced into the protein structure and docked in the pocket of the original ligand. Based on the scores calculated by GBVI/WSA combined with free energy, the top 5 results were analyzed using MOE.

### 4.9. Flow cytometry Analysis of Apoptosis

A549 and DU145 cells (5 × 10^5^ cells/well) were bedded into 6-well plates and treated with **11c** at the indicated concentrations. After 24 h incubation, the cells were washed with phosphate-buffered saline (PBS) twice followed by incubating with the eBioscience™ Annexin VFITC Apoptosis Kit (Invitrogen, City, Carlsbad, California, USA) according to the manufacturer’s instructions. Finally, cell apoptosis was analyzed by flow cytometry and the results were processed by flowjo software(version 10.6.2).

### 4.10. Subcutaneous Tumor Xenograft Model and Antitumor Assay In Vivo

Nude mice were randomly divided into 4 treatment groups (n = 5/group) before being implanted with subcutaneous tumor xenograft cells. DU145 cells were harvested, counted, and reserved on ice. About 15 × 10^6^ cells were injected to establish a tumor transplantation nude mouse model. The four groups were treated as follows: vehicle group, Gefitinib (100 mg/kg) group, and **11c** (5 mg/kg or 10 mg/kg) group. Tumor size (1/2 × length × width^2^) and body weight were measured every 3 days for 21 days. On the last day the mice were sacrificed, and the tumor weight was determined [6].

### 4.11. Immunohistochemical (IHC) Analysis

The xenograft model tumor of mice was collected and fixed with 4% paraformaldehyde (PFA) for 72 h at 4 °C and then embedded in paraffin and cut into slices. After, the slices were baked, dewaxed in xylene, gradients ethanol, and boiled in a microwave to repair antigen [31]. The inactivation of endogenous peroxidase was blocked by incubating with fresh 3% H_2_O_2_. The slices were blocked with fat free milk and incubated with the primary antibody (1:3000) at 4 °C overnight. After that, they were washed with PBS followed by incubation with the HRP-conjugated secondary antibody at room temperature (Boster, Wuhan, China). Finally, slices were dyed by DAB/H_2_O_2_ reaction, brown color in the cell membrane indicated positive staining. Images were captured using an upright fluorescence microscope (Olympus BX53, Tokyo, Japan) [6,52].

### 4.12. Statistical Analysis

The data were presented as mean ± SD. GraphPad prism (version 8.0.2) was used for statistical analysis. *p*-value < 0.05 (*) was considered a significant difference, which was calculated by *t*-test, one and two-way ANOVA (analysis of variance).

## Data Availability

The authors declare that supporting data of this study are available within the article and the Appendix A.

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
