# Peer review of "A Novel Aldisine Derivative Exhibits Potential Antitumor Effects by Targeting JAK/STAT3 Signaling"

_marinedrugs, 2023, doi:10.3390/md21040218_

Round 1
Reviewer 1 Report
The authors investigated a derivative of aldisine from marine sponge as an anti-cancer agent. Chemistry modifications resulted in further analysis of compound 11c as a potential JAK/STAT inhibitor. Cell viability assays showed 11c inhibition at micromolar levels in DU145 and A549 cancer cells. A STAT3 reporter cell line revealed inhibitory activity of 11c. Western blot also showed inhibition of phospho-STAT3 in constitutively active and IL6 inducible cells. Furthermore, western blot showed inhibition of multiple JAKs (1,2, 3, and TyK2), which are the upstream kinases of STAT3. Other signaling pathways were not greatly affected by 11c. Molecular docking showed interactions of 11c with multiple JAK isoforms. Downstream targets of STAT3 were also reduced by 11c; induction of apoptosis was also shown. Finally, in vivo activity of 11c was demonstrated in a DU145 xenograft mouse model.
Overall, results are supportive for 11c as a novel pan-JAK inhibitor with potential anti-cancer activity. There are a few suggestions that are intended to improve the quality of the manuscript.
1 1. Introduction: Ref 1 should cite the most recent report on cancer statistics. Ref 3 journal is missing; similar observation with other citations so needs to be carefully checked. Many refs for Jak (5-7) are older and would like to see newer more up to date citations. Line 46, 47 should remove "clinical trial" and provide specific citation.
2. Fig. 4: Levels of total JAK proteins should be shown to be sure decrease in P-JAK not due to decrease in total JAK. There is some inhibition of NFkB pathway (should comment) as well as increase in p-38/JNK. For this reason, results would also be strengthened if total protein levels are shown.
3. Fig. 7: Are lower tumor weights due to increase in apoptosis? Can try either IHC or western blot using apoptosis marker.
4. Discussion should provide a clear explanation as to the value of pan-JAK inhibitors compared to selective inhibitors. Is the idea that pan inhibitors are more toxic?
5. Discussion: first paragraph sentences need to be revised or corrected.
Reviewer 2 Report
Dong et al.'s manuscript, entitled " A novel aldisine derivative exhibits potential antitumor effects by targeting JAK/STAT3 signaling" studied the Antitumor activity of modified aldisine derivatives. The result suggested that they designed and synthesized a novel small molecule JAKs inhibitor targeting the JAK/STAT3 signaling pathway, which has predicted therapeutic potential for JAK/STAT3 overactivated cancer treatment. However, the author needs to address the following comments before publishing this article in this journal. My comments are
- The author should quantify the protein expression by measuring figures 3, 4 and 6.
- The author needs to clearly show Cell apoptosis results (figure. 6) and label Q1-Q4.
- The author should add statistical analysis for Figure 7A.
Reviewer 3 Report
it appears that the researchers have modified aldisine derivatives to improve their antitumor activity, specifically by introducing an isothiouronium group. They have identified compounds 11a~11c that significantly inhibited JAK/STAT3 activities, and further studies showed that compound 11c exhibited the optimal antiproliferative activity and was a pan-JAKs inhibitor capable of inhibiting constitutive and IL-6-induced STAT3 activation. The researchers also found that compound 11c influenced STAT3 downstream gene expression and induced the apoptosis of A549 and DU145 cells in a dose-dependent manner. Finally, they demonstrated the antiproliferative effects of 11c in an in vivo tumor xenograft experiment with DU145 cells.
Based on this information, it seems that the research described in the abstract could be considered novel and new, as the researchers have modified aldisine derivatives in a novel way and identified a compound (11c) with potent antitumor activity that targets JAK/STAT3 signaling. While there may be other studies that have investigated the use of JAK/STAT3 inhibitors for cancer treatment, the specific compound and modification described in this research may be unique and novel.
Introduction:
The introduction section provides a good overview of the importance of developing new targeted antitumor drugs, especially for cancers that are currently incurable or advanced. However, it would be helpful to include specific examples of such cancers to provide more context for readers.
The description of the JAK/STAT pathway is clear and concise, and the references provided support the statements made. However, it would be helpful to provide a more detailed explanation of how the pathway is involved in cancer development and progression, as well as how JAK inhibitors work to inhibit this pathway.
The description of marine sponges as a potential source of new bioactive natural products is well-supported by references. However, it would be helpful to provide more context on why marine sponges are such a promising source of new drugs, such as their unique adaptations to survive in harsh marine environments.
The description of aldisine and its derivatives is clear and well-supported by references. However, it would be helpful to provide more context on why aldisine is such a promising lead compound for cancer therapeutics, such as its unique structural features or mechanism of action.
The introduction section does a good job of setting up the rationale for the study and the specific compounds tested. However, it would be helpful to provide more details on the specific research question being addressed and how the experiments were designed to answer this question.
There are a few grammatical errors in the introduction section, such as missing articles ("the development of new targeted antitumor drugs is of great significance to human health" should be "the development of new targeted antitumor drugs is of great significance to human health"), inconsistent use of tense ("JAK inhibitors have been shown" vs. "dozens of clinical trials are currently underway"), and missing commas in compound lists ("compound 10a10c 11a11c" should be "compound 10a10c, 11a11c"). These should be corrected to improve the clarity of the writing.
Results
First there is a separate discussion section. So please correct the title.
1. It would be helpful to describe the rationale behind the design of the aldisine derivatives and how they were expected to inhibit JAK/STAT3 signaling.
2. When discussing the synthesis of the compounds, it would be useful to include more details about the reaction conditions and purification methods to allow for reproducibility.
3. It is important to clearly state the experimental procedures used to assess the antiproliferative activity of the compounds.
4. In terms of grammar, some sentences are too long and could be broken up into smaller ones to improve readability. Also, it would be helpful to use active voice instead of passive voice to make the writing more concise and direct.
5. Use consistent verb tense throughout the section.
Discussion:
Scientific comments:
It would be useful to provide more information about the cell lines and animal models used to evaluate the biological activity of compound 11c.
It would be valuable to discuss the potential mechanisms underlying the observed effects of compound 11c on JAK/STAT3 signaling and cancer cell growth in greater detail.
It would be informative to discuss how the findings of this study relate to previous research on JAK inhibitors and their potential clinical applications in cancer therapy.
Grammatical comments:
The sentences in the second and third paragraphs are quite long and complex. Breaking them up into shorter sentences would make the discussion easier to read and follow.
In the third paragraph, the phrase "there is still something therapeutic interests" should be revised to "there are still therapeutic interests".
In the fourth paragraph, the sentence "Compound 11c exhibited inhibition activity" should be revised to "Compound 11c exhibited inhibitory activity".
Round 2
Reviewer 1 Report
Responses to my comments have been addressed in revised version.
Reviewer 2 Report
The authors have satisfactorily responded to all comments and made the necessary changes to the manuscript.
Reviewer 3 Report
I do not have more comments.